# Multi-Target Localization and Tracking Using TDOA and AOA Measurements Based on Gibbs-GLMB Filtering [note 1]

**DOI:** 10.3390/s19245437

**Published:** 2019-12-10

**Authors:** Zhengwang Tian, Weifeng Liu, Xinfeng Ru

**Affiliations:** Xiasha Higher Education Zone, Hangzhou 310018, Zhejiang, China; 17764590774@163.com (Z.T.); a840064210@hdu.edu.cn (X.R.)

**Keywords:** passive localization, time difference of arrival, angle of arrival, random finite sets, Gibbs sampling, GLMB filter, multi-target tracking

## Abstract

This paper deals with mobile multi-target detection and tracking. In the traditional method, there are uncertainties such as misdetection and false alarm in the measurement data, and it will be inevitable having to deal with the data association. To solve the target trajectory and state estimation problem under a cluttered environment, this paper proposes a non-concurrent multi-target acoustic localization tracking method based on the Gibbs-generalized labelled multi-Bernoulli (Gibbs-GLMB) filter and considers an acoustic array of a fixed arrangement for the tracking of targets by joint time difference of arrival (TDOA) and angle of arrival (AOA) measurements. Firstly, the TDOAs are calculated by using the generalized cross-correlation algorithm (GCC) and the AOAs are derived from the received signal directions. Secondly, we assume the independence of the targets and fuse the measurements which are used to track the multiple targets via the Gibbs-GLMB filter. Finally, the effectiveness of the method is verified by Monte Carlo simulation experiments.

## 1. Introduction

Passive detection, such as multi-sensor array localization of acoustic sources, plays an important role in the field of target tracking [1]. Localization through acoustic signals has broad applications in both civil and military fields, for example, the detection of unknown objects in the airport, detection of illegal traffic whistles/horns, localization of submarines or marine animals and the localization of explosion sources. Existing passive localization and tracking techniques include Time Difference of Arrival (TDOA) [2], and Angle of arrival (AOA) [3]. TDOA requires multiple observation devices to be used at the same time to ensure that the clock time of the sensor is consistent among the multiple groups of sensors [4].

In multi-target scenarios, it is difficult to determine the number and states of targets due to clutter effects, miss detection and data association uncertainty. Reference [5] proposed the route-based dynamic modeling to improve data association performance. For the complex and non-linear acoustic signals, traditional signal processing techniques such as the Fang algorithm, the music algorithm and the Taylor series expansion method are used [2]. To address the nonlinearity of the measurement process, the extended Kalman filter, one of the common methods in target tracking, can be used to provide better information for multi-sensor information fusion. In Reference [6], they performed data fusion that combines the active detection and the passive interception using Maximum Likelihood Estimation (MLE). In Reference [7], MLE based on compressed sensing is proposed for the TDOA method. In Reference [8], the authors developed MLE for the proposed model through the Gauss-Newton iteration and semidefinite relaxation. An extended Kalman particle filtering (EKPF) approach for non-concurrent multiple acoustic tracking (NMAT) has been studied in Reference [9]; however, this paper only considers single scan tracking.

Currently, The Joint Probabilistic Data Association Filter (JPDAF), Multiple Hypothesis Tracking (MHT) and Random Finite Set (RFS) [10] are three mainstream methods for multi-target tracking. Traditional approaches such as JPDAF and MHT are extended to solve multi-target tracking based on single-target tracking, mostly in an ad-hoc manner. As the number of targets increases, the calculation amount of these methods will increase exponentially. The RFS approach pioneered by Mahler [11,12,13,14] provides a top-down state-space model formulation for multiple object system based on fundamental concepts in estimation theory, such as multi-target estimation error [15] and Bayes optimality [13,14]. Due to its mathematically rigorous foundation, the RFS theory has received worldwide attention in recent years and is considered to be a way to solve multi-target tracking.

Many well-known multi-target filters have been developed from the RFS framework, for example, the Probability Hypothesis Density filter [11,16,17], cardinality-banlance multi-target multi-Bernoulli filter [18], Cardinalized PHD filter [12,19]. However, these filters can only obtain the scatter set estimation of the target but cannot form the target trajectories, though several heuristics have been proposed to join state estimates from different times steps to form trajectories. In spite of this, these filters have been widely used in many fields, for example, computer vision [20,21,22]; sensor scheduling [23,24]; multi-sensor fusion [25]. Reference [26] propose to solve the multi-target sensor management by using the random set method in the POMDP ] framework; References [27,28,29] use Cauchy-Schwarz divergence and Rényi divergence as information gains, respectively, provide a new sensor scheduling; robotics [30,31] and group target tracking [32].

The RFS multi-target trackers are formulated via labeled RFS [13,14,33]. In Reference [33], Mahler has shown that labeled RFS is the only principled approach that can provide target trajectories from the filtering density. The recent breakthrough in multi-target tracking is a filter labeled RFS called the Generalized Labeled Multi-Bernoulli (GLMB) or the Vo-Vo filter [34,35]. This filter is the first analytic method to Bayes filter with the multi-target, which provides estimates of target trajectories with linear complexity, and can be efficiently implemented by jointing the update step and the prediction step, for more effective multi-target tracking [36]. GLMB filtering has been demonstrated to track more than one million targets in heavy clutter, misdetections and data association uncertainty [37]. Another advantage of labeled RFS over unlabeled RFS is that it can provide ancestry or lineage information in problems that involve spawning targets [38]. Such capability is not possible without labels.

GLMB RFS has been applied in many fields, such as tracking with merged measurements [39], extended targets [40], computer vision [41,42,43], cell tracking [44,45], track-before-detect [46,47], sensor scheduling [48,49], field robotics [50,51,52], distributed tracking [53,54] and cell microscopy [55]. The GLMB solution has also been applied to the multi-sensor case [56] and the multi-scan case [57]. The multi-sensor GLMB filter [56] is the first multi-sensor solution with linear complexity in the sum of number of measurements. The multi-scan GLMB filter [57] is the first solution that is demonstrated to handle as much as 100 scans as well as providing posterior statistics about the set of target trajectories.

The work in this paper is based on Reference [1] and the published conference papers [58]. The measurement data is calculated from the real sound source signal, and the positioning and tracking of multiple target states via the GLMB filter under the RFS framework. To implement the GLMB filter more effectively, Vo et al. proposed joint the update step and the prediction step, eliminating the inefficiency caused by the primal two process redundancy and adopting the Gibbs sampling method in the truncation process [36], which provide a more effective solution for the ranked assignment problem(the data association problems). The Gibbs-GLMB has been proven to provide faster and more accurate results based on the GLMB filter.

Multi-acoustic array localization is typical of multi-sensor, passive localization and nonlinear problems. In the case of traditional algorithms dealing with multi-sensor measurement uncertainty and target uncertainty, the target tracking is based on the method of associating data and there is no effective way to estimate the number of targets. The Gibbs-GLMB filtering can effectively solve these problems. We change from the original active tracking to passive tracking based on the original Gibbs-GLMB filter [36]. TDOA and AOA measurements are generated by calculating the true signal correlation, which is a nonlinear estimation problem. The TDOAs and the AOAs are computed by the generalized cross-correlation (GCC) [4] and receiving direction of acoustic signal, respectively.

For the reasons above, we use the Gibbs-GLMB filter. First, we assume that the target obeys a cv motion model and the observation information includes TDOA and AOA. Secondly, target tracking in a multi-sensor array is done using the Gibbs sampling implementation of the GLMB filter (Gibbs-GLMB filter), to reduce the computational complexity of the algorithm without sacrificing accuracy. The effectiveness of the algorithm is verified by three pairs of acoustic array sensors are deployed to track three targets.

## 2. Background

### 2.1. NOTATION

Single-target state is expressed by a small letter, (e.g., *x*).Multi-target states are represented by an italic capital letter, (e.g., *X*).The labeled states and distribution are bolded, (e.g., x,X,π).The spaces are represented by blackboard bold (e.g., the state space X and measurement space Z).$F\left(X\right)$ is the all finite subsets of X.The inner product symbol is abbreviated as:$\langle f,g\rangle ≜\int f\left(x\right)g\left(x\right)dx$.The following multi-target exponential notation $h^{X}≜\prod_{x\in X}h\left(x\right)$, where *h* is a real-valued function, with $h^{\emptyset }\;=\;1$.The generalization of the Kronecker delta for sets, vectors and integers:
$$\begin{array}{c} \delta_{Y}\left(X\right)=\left\{\begin{array}{c} 1,ifX=Y\\ 0,otherwise, \end{array}\right. \end{array}$$
where inclusion function is denoted as:
$$\begin{array}{c} 1_{Y}\left(X\right)=\left\{\begin{array}{c} 1,ifX\subseteq Y\\ 0,otherwise. \end{array}\right. \end{array}$$$X_{m:n}$ is shorthand for the list of variables $X_{m},X_{m+1},\cdots ,X_{n}$.

### 2.2. Random Finite Set

In the multi-target environment based on RFS framework in time k, the states of multiple targets can be denoted by a set as $X_{k}=\left\{x_{k,1},\cdots ,x_{k,N\left(k\right)}\right\}\in F\left(X\right)\;$ [13], where $F\left(X\right)$ is the all of finite subsets of state space and $N\left(k\right)$ is the number of surviving targets. In the similar way, the observation about TDOAs and AOAs of the *q*th sensor pair can be described as $Z_{k}^{\left[q\right]}=\left\{z_{k,1}^{\left[q\right]},\cdots ,z_{k,M_{q}\left(k\right)}^{\left[q\right]}\right\}\in F\left(Z\right)$, where $F\left(Z\right)$ is the space of finite subsets of observation space Z and $M_{q}\left(k\right)$ is the number of observed measurement.

In the location tracking area, there are many uncertainties in the number and measurement of the detection process, such as birth, death, derivation, false alarm and missed detection. Consequently, The multiple targets state in time *k* can be defined as [59,60]:
(1)$$X_{k}=\left[\underset{x\in X_{k-1}}{⋃}S_{k\left|k-1\right.}\left(x\right)\right]\cup \left[\underset{x\in X_{k-1}}{⋃}B_{k\left|k-1\right.}\left(x\right)\right]⋃\Gamma_{k},$$
where $S_{k\left|k-1\right.}\left(x\right)$, $B_{k\left|k-1\right.}\left(x\right)$ and $\Gamma_{k}$ are the RFS of survival target at time k−1, the RFS of the target spawn at time *k* from the survival target at time k−1 and the RFS of target new-born, respectively. There are clutter or false alarms in the tracking area, which can be expressed as:(2)$$Z_{k}^{\left[q\right]}=\left[\underset{x\in X_{k}}{⋃}\Theta_{k}^{\left[q\right]}\left(x\right)\right]⋃K_{k}^{\left[q\right]},$$
where $\Theta_{k}^{\left[q\right]}\left(x\right)$ is the measurements with the RFS which produced by targets in the tracking area:(3)$$\Theta_{k}^{\left[q\right]}\left(x_{k}\right)\;=\;\left\{\begin{array}{c} \phi ,\;\;\;H_{miss}\\ \left\{z_{k}^{\left[q\right]}\right\},{\bar{H}}_{miss} \end{array}\right.$$
here, the ${\bar{H}}_{miss}$ and $H_{miss}$ are the hypotheses of detection and miss detection, more generally, whether the sensor has received the signal generated by targets. Moreover, $K_{k}^{\left[q\right]}$ is the measurement set of alarms or clutter false which follows a poisson distribution with a uniform density $U\left(z\right)$ on the observation area and is given by:(4)$$K_{k}\left(z_{k}\right)=\frac{\lambda_{c}}{\int U\left(z\right)dz}U\left(z\right).$$

In RFS framework, the probability density function that the state of multi-target makes a transition from state $X_{k-1}$ to $X_{k}$ can be described as:(5)$$\begin{array}{c} f_{k\left|k-1\right.}\left(X_{k}\left|X_{k-1}\right.\right)=\sum_{W\in X_{k}}\pi_{T,k\left|k-1\right.}\left(W\left|X_{k-1}\right.\right)\times \pi_{\Gamma ,k}\left(X_{k}-W\right), \end{array}$$
where $\pi_{T,k\left|k-1\right.}\left(\cdot \left|\cdot \right.\right)$ is the probability density of spontaneous target birth and $\pi_{\Gamma ,k}\left(\cdot \right)$ is the probability density of target new-born.

### 2.3. Multi-Bernoulli RFS

The state of the target and the measurements are random variables. The Bernoulli distribution can be used to describe a single target X∈X. Hence, a singleton target probability *X* is *r* which satisfies the spatial distribution of a probability density $p\left(x\right)$ and the probability that the target does not exist is 1−r. The probability density distribution of the Bernoulli RFS is written as follows:(6)$$\pi \left(x\right)\;=\;\left\{\begin{array}{c} 1-r\begin{array}{cc} \;\;\; & \;X\;=\;\emptyset \end{array}\\ r\cdot p\left(x\right)\begin{array}{cc} \; & X=\left\{x\right\} \end{array}\\ 0\begin{array}{ccc} \; & \; & \mathrm{otherwise}. \end{array} \end{array}\right.$$

The Multi-Bernoulli RFS is given by combining of the *M* independent Bernoulli RFS $X^{\left(i\right)}\in X,i=1,\cdots ,M$(satisfying $X=\overset{M}{\underset{i=1}{⋃}}X^{\left(i\right)}$) with existence probability $r^{\left(i\right)}\in \left(0,1\right)$, which is described by ${\left\{\left(r^{\left(i\right)},p^{\left(i\right)}\right)\right\}}_{i=1}^{M}$. $\sum_{i\;=\;1}^{M}r^{\left(i\right)}$ is the mean cardinality of the Multi-Bernoulli RFS. Therefore, the probability density distribution of multi-Bernoulli is expressed by [12,18]:(7)$$\pi \left(\left\{x_{1},\ldots ,x_{n}\right\}\right)=\prod_{j=1}^{M}\left(1-r^{\left(j\right)}\right)\sum_{1⩽i_{1}\neq \cdots \neq i_{n}⩽M}\prod_{j=1}^{n}\frac{r^{\left(i_{j}\right)}p^{\left(i_{j}\right)}\left(x_{j}\right)}{1-r^{\left(i_{j}\right)}}.$$

### 2.4. Labeled Multi-Bernoulli RFS

A labeled-random finite set (L-RFS) [34,35] means that each state of the RFS has a unique tag. This means we attach a unique label $l\in L=\left\{\alpha_{i}:i\in N\right\}$ to each state x∈X where L is discrete countable space and N is the positive integer set space. The single target state is expressed as:(8)$$x_{k,N\left(k\right)}\;=\;\left(x_{k,N\left(k\right)},l_{k,N\left(k\right)}\right)\in X\times L.$$

The labels of the set X⊂X×L can be represented by $L\left(X\right)=\left\{L\left(x\right):x\in X\right\}$, where L:X×L→L is defined by $L\left(\left(x,l\right)\right)=l$. The distinct label indicator is defined by $\Delta \left(X\right)=\delta_{\left|X\right|}\left(\left|L\left(X\right)\right|\right)$.

The parameter of a labeled multi-Bernoulli(LMB) RFS can be described as a set $\left\{\left(r^{\left(\zeta \right)},p^{\left(\zeta \right)}\right):\zeta \in \Psi \right\}$ with index set Ψ. We extend the problem on space X to space X×L, thus, the probability density distribution of labeled LMB-RFS is given by [34]:(9)$$\begin{array}{c} \pi \left(\left\{\left(x_{1},l_{1}\right),\ldots \left(x_{n},l_{n}\right)\right\}\right)=\delta_{n}\left(\left|\left\{l_{1},\ldots l_{n}\right\}\right|\right)\prod_{\zeta \in \Psi }\left(1-r^{\left(\zeta \right)}\right)\prod_{j=1}^{n}\frac{1_{\alpha \left(\Psi \right)}\left(l_{j}\right)r^{\left(\alpha^{-1}\left(l_{j}\right)\right)}p^{\left(\alpha^{-1}\left(l_{j}\right)\right)}\left(x_{j}\right)}{1-r^{\left(\alpha^{-1}\left(l_{j}\right).\right)}} \end{array}$$

The following simplified alternative form of the LMB can be simplified as:(10)$$\pi \left(X\right)=\Delta \left(X\right)1_{\alpha \left(\Psi \right)}\left(L\left(X\right)\right){\left[p\left(\cdot \right)\right]}^{X}$$

### 2.5. GLMB RFS

A generalized label multi-Bernoulli RFS under the state space X and the label space L has the following distribution [34]:(11)$$\pi \left(X\right)=\Delta \left(X\right)\sum_{\xi \in \Xi }\omega^{\left(\xi \right)}\left(L\left(X\right)\right){\left[p^{\left(\xi \right)}\right]}^{X},$$
where $\xi =\left(\theta_{1:k}\right)\in \Theta$ is a historical association maps. The non-negative $\omega^{\left(\xi \right)}\left(L\right)$ and a probability density $p^{\left(\xi \right)}$ satisfy: (12)$$\sum_{L\in L}\sum_{\xi \in \Xi }\omega^{\left(\xi \right)}\left(L\right)=1$$
(13)$$\int p^{\left(\xi \right)}\left(x,l\right)dx=1.$$

## 3. Problem Formulation

### 3.1. Model Environment

There are multiple sets of acoustic sensor arrays in the detection range, denoted as $S_{1:Q}\;=\;\left\{s_{1},\cdots ,s_{q},\cdots ,s_{Q}\right\},q\in \left\{1,2,\cdots ,Q\right\}$, where $s_{q}\;=\;\left\{s_{q,1},\cdots ,s_{q,j},\cdots ,s_{q,N}\right\},j\in \left\{1,\cdots ,N\right\}$. Each $s_{q,j}$ can be defined as $s_{q,j}\;=\;\left(x_{q,j},y_{q,j}\right)$, in 2-dimensional space.

Assuming a single target state of position is $x_{i}=\left(x_{i},y_{i}\right)$, each set of sensors consists of two acoustic sensors.

#### 3.1.1. Time Difference of Arrival

The time difference τ is expressed as:(14)$$TDOA_{q}\;=\;\left|\frac{∥x_{i}-s_{q,1}∥-∥x_{i}-s_{q,2}∥}{v}\right|,$$
where both $x_{i}$ and $s_{q,j}$ are in Cartesian coordinates, $∥\cdot ∥$ is the Euclidean-norm and *v* is the velocity of sound.

#### 3.1.2. Angle of Arrival

For sensor 1, AOA can be expressed as:(15)$$AOA_{q,1}\;=\;\arctan\left(\frac{y_{i}-y_{q,1}}{x_{i}-x_{q,1}}\right).$$

For ease of understanding, the TDOA and AOA are illustrated in the positioning system of Figure 1, where: $TDOA=\left|\frac{MN}{velocity\;of\;sound\;}\right|$, AOA=α.

### 3.2. Measurement

#### 3.2.1. TDOA Measurement

The signals observed by a pair of sensors can be mathematically described as:(16)$$z_{1}\left(t\right)=\alpha_{1}s\left(t\right)+n_{1}\left(t\right)$$
(17)$$z_{2}\left(t\right)=\alpha_{2}s\left(t-\tau_{1:2}\right)+n_{2}\left(t\right),$$
where $z_{1}\left(t\right)$ and $z_{2}\left(t\right)$ are the signals received by the pair of sensor array, $s\left(t\right)$ is the true signal, $n_{1}\left(t\right)$ and $n_{2}\left(t\right)$ are noise signals, $\tau_{1:2}$ is the time difference between two sensors detecting the signal, $\alpha_{1}$ and $\alpha_{2}$ are signal amplitudes [8].

The time difference can be estimated by the generalized cross correlation (GCC) method [4]:(18)$$R_{GCC}\left(\tau_{q}\right)\;=\;\int_{-\infty }^{\infty }\psi_{12}\left(\omega \right)Z_{1}\left(\omega \right)Z_{2}^{*}\left(\omega \right)e^{-j\omega \tau_{q}}d\omega$$
(19)$${\hat{\tau }}_{q}=\arg\max R_{GCC}\left(\tau_{q}\right)$$
(20)$$\psi_{1,2}\left(\omega \right)\;=\;\frac{1}{\left|G_{x_{1}x_{2}}\left(\omega \right)\right|}\;=\;\frac{1}{\left|Z_{1}\left(\omega \right)Z_{2}^{*}\left(\omega \right)\right|}.$$

Here, $R_{GCC}\left(\tau_{q}\right)$ is the GCC, where $Z_{1}\left(\omega \right)$ is the Fourier transforms of $z_{1}\left(t\right)$ and $Z_{2}^{*}\left(\omega \right)$ is the Fourier transform conjugate of $z_{2}\left(t\right)$. $\psi_{1,2}\left(\omega \right)$ is the weight function of GCC. To reduce environmental noise and reverberation interference, we choose the phase transform (PHAT) as our weight function, the formula is given by $\psi_{1,2}\left(\omega \right)\;=\;\frac{1}{\left|G_{x_{1}x_{2}}\left(\omega \right)\right|}$.

#### 3.2.2. AOA Measurement

The AOA is calculated by the positional relationship between the sensor and the target position. The difference in the AOAs of the sensor array is calculated by combining the measurements of the two sensors:
(21)$$\delta_{q}\;=\;\left|\arctan\left(\frac{y_{i}-y_{q,1}}{x_{i}-x_{q,1}}\right)-\arctan\left(\frac{y_{i}-y_{q,2}}{x_{i}-x_{q,2}}\right)\right|.$$

### 3.3. Motion Model

We take the CV model as an example of a linear model, also known as a non-maneuver model:(22)$$\left[\begin{array}{c} \dot{x}\\ \ddot{x} \end{array}\right]\;=\;\left[\begin{array}{cc} 0 & 1\\ 0 & 0 \end{array}\right]\left[\begin{array}{c} x\\ \dot{x} \end{array}\right]+\left[\begin{array}{c} 0\\ 1 \end{array}\right]w\left(t\right),$$
where *x* is the location of the target, $\dot{x}$ is the velocity of the target, $\ddot{x}$ is the acceleration of the target, $w\left(t\right)$ is zero mean white noise. Let *T* denotes the sampling interval, then the discrete-time model is given by:(23)xk+1x˙k+1=1T01xkx˙k+T2T222Twt.

## 4. TDOA Localization Algorithm Based on SMC-GLMB Filtering

### 4.1. Target State Estimation

The purpose of multi-target tracking is to jointly estimate the target cardinality and target states based on the observations. Multi-target tracking can be transformed into the recursive Bayesian estimation problem by modeling the state and measurement of multi-target using RFS. The RFS approach can effectively deal with the uncertainty of data association between the target and the measurement and the state probability density function of the set of targets. We use $\pi_{k}\left(\left.\cdot \right|Z_{1:k}\right)$ to indicate the RFS posterior probability density of multi-target state; $f_{k\left|k-1\right.}\left(\cdot \left|\cdot \right.\right)$ to represent the multi-target transition density; $g_{k}\left(\cdot \left|\cdot \right.\right)$ to represent the likelihood function. The posterior probability density of multiple targets is recursively calculated by the following prediction and update steps [11,12,13,14]:$$\begin{array}{c} \pi_{k\left|k-1\right.}\left(X_{k}\left|Z_{1:k-1}\right.\right)=\int f_{k\left|k-1\right.}\left(X_{k}\left|X_{k-1}\right.\right)\pi_{k-1}\left(X_{k-1}\left|Z_{1:k-1}\right.\right)\delta X_{k-1} \end{array}$$
$$\pi_{k}\left(X_{k}\left|Z_{1:k}\right.\right)=\frac{g_{k}\left(Z_{k}\left|X_{k}\right.\right)\pi_{k\left|k-1\right.}\left(X_{k}\left|Z_{1:k-1}\right.\right)}{\int g_{k}\left(Z_{k}\left|X_{k}\right.\right)\pi_{k\left|k-1\right.}\left(X_{k}\left|Z_{1:k-1}\right.\right)\delta X_{k}},$$
where the set integral on $F\left(X\times L\right)\to R$ is defined as:$$\int f\left(X\right)\delta X=\sum_{i=0}^{\infty }\frac{1}{i!}\int f\left(\left\{x_{1},\cdots ,x_{i}\right\}\right)d\left(x_{1},\cdots ,x_{i}\right).$$

All information about the multiple targets states are included in the multi-target posterior, for example, the number and state of the target at the current time.

We experience with *q* pairs of sensors, therefore the above update step and prediction step can be rewritten as [59,60,61]:(24)$$\pi_{k\left|k-1\right.}\left(X_{k}\left|Z_{1:k-1}^{\left[1:Q\right]}\right.\right)=\int f_{k\left|k-1\right.}\left(X_{k}\left|X_{k-1}\right.\right)\pi_{k-1}\left(X_{k-1}\left|Z_{1:k-1}^{\left[1:Q\right]}\right.\right)\delta X_{k-1}$$
(25)$$\pi_{k}\left(X_{k}\left|Z_{1:k-1}^{\left[1:Q\right]}\right.\right)=\frac{\prod_{q=1}^{Q}g_{k}\left(Z_{k}^{\left[q\right]}\left|X_{k}\right.\right)\pi_{k\left|k-1\right.}\left(X_{k}\left|Z_{1:k-1}^{\left[1:Q\right]}\right.\right)}{\int \prod_{q=1}^{Q}g_{k}\left(Z_{k}^{\left[q\right]}\left|X_{k}\right.\right)\pi_{k\left|k-1\right.}\left(X_{k}\left|Z_{1:k-1}^{\left[1:Q\right]}\right.\right)\delta X_{k}}.$$

The recursive process is not easy to deal with exactly due to the non-linear of the observation equation. The sequential Monte Carlo(SMC) methods are a viable approach to approximate the integrals of interest using random samples.

### 4.2. Particle Filter Implementation

Since the multi-target posterior probability density recursion requires the calculation of multi-set integral (Equation 24) and (Equation 25), its computational complexity is much larger than that of the single-target filtering process [16,61,62]. By the SMC method, the weighted particles can be estimated by recursive approximation to estimate the posterior probability density.

At the current time *k*, the particles are sampled by SMC, obtained from the spatial distribution of the target.

(26)$${\tilde{X}}_{k}^{\left(i\right)}∼p\left(\cdot \left|X_{k-1}^{\left(i\right)},Z_{k}\right.\right)$$

${\left\{\omega_{k-1}^{\left(i\right)},X_{k-1}^{\left(i\right)}\right\}}_{i=1}^{N}$ represents the set of importance weighted particles at time k−1 and the multi-target posterior probability density can be expressed as:(27)$$\pi_{k-1\left|k-1\right.}\left(X_{k}\left|Z_{1:k-1}\right.\right)\approx \sum_{i=1}^{N}\omega_{k-1}^{\left(i\right)}\delta_{X_{k-1}^{\left(i\right)}}\left(X_{k-1}\right)$$

**Algorithm 1** Particle Filter
1:
**for**
k=1,⋯,T
**do**
2: **for**
i=1,⋯,N
**do**3:  Sample ${\tilde{X}}_{k}^{\left(i\right)}∼p\left(\cdot \left|X_{k-1}^{\left(i\right)},Z_{k}\right.\right)$;4:  Set ${\tilde{\omega }}_{k}^{\left(i\right)}=\omega_{k-1}^{\left(i\right)}\frac{g_{k}\left(Z_{k}\left|{\tilde{X}}_{k}^{\left(i\right)}\right.\right)f_{k\left|k-1\right.}\left({\tilde{X}}_{k}^{\left(i\right)}\left|X_{k-1}^{\left(i\right)}\right.\right)}{p_{k\left|k-1\right.}\left({\tilde{X}}_{k}^{\left(i\right)}\left|X_{1:k-1}^{\left(i\right)},Z_{1:k}\right.\right)}$;5:  Normalise weights $\omega_{k}^{\left(i\right)}=\frac{{\tilde{\omega }}_{k}^{\left(i\right)}}{\sum_{j=1}^{N}{\tilde{\omega }}_{k}^{\left(j\right)}}$, here $\sum_{i=1}^{N}\omega_{k}^{\left(i\right)}\;=\;1$;6: **end for**7:
**end for**
8:Resample ${\left\{\omega_{k}^{\left(i\right)},X_{k}^{\left(i\right)}\right\}}_{i=1}^{N}$ and get ${\left\{{\tilde{\omega }}_{k}^{\left(i\right)},{\tilde{X}}_{k}^{\left(i\right)}\right\}}_{i=1}^{N}$;9:Set ${\hat{\pi }}_{k}=\sum_{i=1}^{N}\omega_{k}^{\left(i\right)}\delta_{X_{k}^{\left(i\right)}}$ as the estimated posterior probability density;


### 4.3. The Multi-Sensor Likelihood

Given the multi-target state *X*, each $\left(x,l\right)\in X$ is either detected with probability $p_{D,m}\left(x,l\right)$ and generates observation *z* with likelihood function $g\left(z\left|x,l\right.\right)$. For *S* sensors, the multi-sensor and multi-target mapping [56] is defined by $\theta^{\left(m\right)}:L\to \left\{0,1\cdots ,\left|Z^{\left(m\right)}\right|\right\}$, m=1,…,S. The set Θ represents the space of vector maps $\theta =(\theta^{\left(1\right)},\ldots ,\theta^{\left(S\right)})$. Assuming that the target and clutter generation are independent and the multi-sensor likelihood function is given by [34]:$$g\left(Z|X\right)\propto \sum_{\theta \in \Theta \left(L\left(X\right)\right)}{\left[\psi_{Z_{1}}\left(\cdot ;\theta^{\left(1\right)}\right)\right]}^{X}\cdots {\left[\psi_{Z_{m}}\left(\cdot ;\theta^{\left(S\right)}\right)\right]}^{X}$$
$$\psi_{Z_{m}}\left(x,l;\theta \right)=\left\{\begin{array}{c} \frac{p_{D,m}\left(x,l\right)g\left(z_{\theta \left(l,m\right)}|x,l\right)}{K_{m}\left(z_{\theta \left(l,m\right)}\right)},\;\;\mathrm{if}\;\theta \left(l,m\right)>0\\ 1-p_{D,m}\left(x,l\right),\;\;\;\;\;\;\;\;\;\;\mathrm{if}\;\theta \left(l,m\right)=0 \end{array}\right.,$$
where $p_{D,m}\left(x,l\right)$ is the probability detection, $K_{m}$ is Poisson clutter, for sensor *m*.

### 4.4. GLMB Filter

The GLMB filter is a Bayesian recursion from the multi-Bernoulli distribution, which satisfies the following formula [34]:(28)$$\begin{array}{c} C=F\left(L\right)\times \Xi \\ \omega^{\left(c\right)}\left(L\right)=\omega^{\left(I,\xi \right)}\left(L\right)=\omega^{\left(I,\xi \right)}\delta_{I}\left(L\right)\\ p^{\left(c\right)}=p^{\left(I,\xi \right)}=p^{\left(\xi \right)}. \end{array}$$

The forward propagation expression of GLMB Filter is as follows:(29)$$\pi \left(X\right)\;=\;\Delta \left(X\right)\sum_{\left(I,\xi \right)\in F\left(L\right)\times \Xi }\omega^{\left(I,\xi \right)}\delta_{I}\left(L\left(X\right)\right){\left[p^{\left(\xi \right)}\right]}^{X}.$$

The distribution of multi-target prior probability is given by the Equation (Equation 29), thus, the multi-target prediction is still the multi-Bernoulli distribution and the prediction step can be expressed as:(30)$$\pi_{+}\left(X_{+}\right)=\Delta \left(X_{\;+\;}\right)\sum_{\left(I_{\;+\;},\xi \right)\in F\left(L\right)\times \Xi }{\omega_{\;+\;}}^{\left(I_{\;+\;},\xi \right)}\delta_{I_{\;+\;}}\left(L\left(X_{\;+\;}\right)\right){\left[{p_{\;+\;}}^{\left(\xi \right)}\right]}^{X^{\;+\;}}$$
where
(31)$${\omega_{+}}^{\left(I_{+},\xi \right)}={\omega_{s}}^{\left(\xi \right)}\left(I_{+}\cap L\right)\omega_{B}\left(I_{+}\cap B\right)$$
(32)$${p_{+}}^{\left(\xi \right)}\left(x,l\right)=1_{L}\left(l\right)p_{s}^{\left(\xi \right)}\left(x,l\right)+\left(1-1_{L}\left(l\right)\right)p_{B}\left(x,l\right)$$
(33)$$p_{s}^{\left(\xi \right)}\left(x,l\right)=\frac{\langle p_{s}\left(\cdot ,l\right)f\left(x|\cdot ,l\right),p^{\left(\xi \right)}\left(\cdot ,l\right)\rangle }{\eta_{s}^{\left(\xi \right)}\left(l\right)}$$
(34)$$\eta_{s}^{\left(\xi \right)}\left(l\right)=\int \langle p_{s}\left(\cdot ,l\right)f\left(x|\cdot ,l\right),p^{\left(\xi \right)}\left(\cdot ,l\right)\rangle dx$$
(35)$${\omega_{s}}^{\left(\xi \right)}\left(L\right)={\left[\eta_{s}^{\left(\xi \right)}\right]}^{L}\sum_{I\in L}1_{I}\left(L\right){\left[{q_{s}}^{\left(\xi \right)}\right]}^{I-L}\omega^{\left(I,\xi \right)}$$
(36)$${q_{s}}^{\left(\xi \right)}\left(l\right)=\langle q_{s}\left(\cdot ,l\right),p_{s}^{\left(\xi \right)}\left(\cdot ,l\right)\rangle$$

Here, the $\omega_{B}\left(I_{+}\cap B\right)$ and ${\omega_{s}}^{\left(\xi \right)}\left(I_{+}\cap L\right)$ are weights of the birth labels $\left(I_{+}\cap B\right)$ and surviving labels $\left(I_{+}\cap L\right)$, respectively. $p_{B}\left(x,l\right)$ is density of a new-born target, $p_{s}^{\left(\xi \right)}\left(x,l\right)$ is the probability density of the surviving target obtained from the prior probability $p^{\left(\xi \right)}\left(\cdot ,l\right)$. $f\left(x|\cdot ,l\right)$ means density weighted by the probability of survival $p_{s}\left(\cdot ,l\right)$.

Given the predicted density as Equation (Equation 29), the update step can be expressed in the form of a truncated estimate:(37)$$\pi \left(X\left|Z\right.\right)\approx \Delta \left(X\right)\sum_{\left(I,\xi \right)\in F\left(L\right)\times \Xi }\sum_{\theta \in \Theta^{\left(M\right)}}{\tilde{\omega }}^{\left(I,\xi ,\theta \right)}\left(Z\right)\delta_{I}\left(L\left(X\right)\right){\left[p^{\left(\xi ,\theta \right)}\left(\left.\cdot \right|Z\right)\right]}^{X},$$
where $\left(I,\xi \right)$ is fixed parameter. For *M* elements set $\Theta^{\left(M\right)}\;=\;\left\{\xi^{\left(1\right)},\cdots ,\xi^{\left(M\right)}\right\}$ there is the highest weight $\omega^{\left(I,\xi ,\theta \right)}$, ${\tilde{\omega }}^{\left(I,\xi ,\theta \right)}$ is the re-normalized weight after truncation and
(38)$${\tilde{\omega }}^{\left(I,\xi ,\theta \right)}\left(Z\right)=\frac{\delta_{\theta^{-1}\left(\left\{0:\left|Z\right.\right\}\right)}\left(I\right)\omega^{\left(I,\xi \right)}{\left[\eta_{Z}^{\left(\xi ,\theta \right)}\right]}^{I}}{\sum_{\left(I,\xi \right)\in F\left(L\right)\times \Xi }\sum_{\theta \in \Theta }\delta_{\theta^{-1}\left(\left\{0:\left|Z\right.\right\}\right)}\left(I\right)\omega^{\left(I,\xi \right)}{\left[\eta_{Z}^{\left(\xi ,\theta \right)}\right]}^{I}}$$
(39)$$p^{\left(\xi ,\theta \right)}\left(x,l\left|Z\right.\right)=\frac{p^{\left(\xi \right)}\left(x,l\right)\psi_{Z}\left(x,l;\theta \right)}{\eta_{Z}^{\left(\xi ,\theta \right)}\left(l\right)}$$
(40)$$\eta_{Z}^{\left(\xi ,\theta \right)}\left(l\right)\;=\;\langle p^{\left(\xi \right)}\left(\cdot ,l\right)\psi_{Z}\left(\cdot ,l;\theta \right)\rangle$$
(41)$$\psi_{Z}\left(x,l;\theta \right)\;=\;\delta_{0}\left(\theta \left(l\right)\right)q_{D}\left(x,l\right)+\left(1-\delta_{0}\left(\theta \left(l\right)\right)\right)\frac{p_{D}\left(x,l\right)g\left(z_{\theta \left(l\right)}|x,l\right)}{K\left(z_{\theta \left(l\right)}\right)}.$$

### 4.5. Gibbs-GLMB Filter

Gibbs sampling is a special case of continuous Markov Chain Monte Carlo (MCMC), which can transform sampling from high-dimensional space to low-dimensional one [63]. Assuming that the target state is $X_{k}=\left\{x_{k,1},\cdots ,x_{k,N\left(k\right)}\right\}$, which obeys the probability distribution π, the probability distribution of the target state is $\pi \left(\left\{x_{k,1},\cdots ,x_{k,N\left(k\right)}\right\}\right)$.

**Algorithm 2** Gibbs sampling
1:
**for**
k=1:T
**do**
2: **for**
n=1:N(k)
**do**3:  $x_{k,n}∼\pi_{n}\left(\cdot \left|x_{k,1:n-1},x_{k-1,n+1:N\left(k\right)}\right.\right)$;4: **end for**5: $X_{k}=\left\{x_{k,1},\cdots ,x_{k,N\left(k\right)}\right\}$;6:
**end for**



In Algorithm 2, $x_{k,1:n-1}$ is the samples $x_{k,1},\cdots ,x_{k,n1}$ that have generated at current time, $x_{k-1,n+1:N\left(k\right)}$ is associations $x_{k,n+1},\cdots ,x_{k,N\left(k\right)}$ at previous time. The Gibbs sampling algorithm reduces the joint density estimation problem to conditional probability to reduce the sampling difficulty and finally updates all parameters by the iterative process of each parameter.

In the calculation process of the update step and the prediction step of the GLMB filter, the number of weights and data quantities of the update and the prediction step are exponentially increasing. By using optimal assignment implementation and the kth shortest path, the complexity of the measurement quantity is cubic [35]. The GLMB filter is truncated by Gibbs sampling, thereby joint prediction and update reduce the complexity of the measurements to linear.

Given the GLMB distribution (Equation 11) at the current time, the GLMB distribution at the next time is [36]:(42)$$\pi_{Z_{\;+\;}}\left(X\right)\propto \Delta \left(X\right)\sum_{I,\xi ,I_{+},\theta_{+}}\omega^{\left(I,\xi \right)}\omega_{Z_{\;+\;}}^{\left(I,\xi ,I_{+},\theta_{+}\right)}\delta_{I_{+}}\left(L\left(X\right)\right){\left[p_{Z_{\;+\;}}^{\left(\xi ,\theta_{+}\right)}\right]}^{X}\;$$
where $I\in F\left(L\right)$, ξ∈Ξ, $I_{\;+\;}\in F\left(L_{\;+\;}\right)$, $\theta_{\;+\;}\in \Theta_{+}$ and
(43)$$\omega_{Z_{\;+\;}}^{\left(I,\xi ,I_{+},\theta_{+}\right)}\;=\;1_{\Theta_{+}\left(I_{+}\right)}\left(\theta_{+}\right){\left[1-{\bar{P}}_{S}^{\left(\xi \right)}\right]}^{I-I_{+}}{\left[{\bar{P}}_{S}^{\left(\xi \right)}\right]}^{I\cap I_{+}}{\left[1-r_{B,+}\right]}^{B_{+}-I_{+}}r_{B,+}^{B_{+}\cap I_{+}}{\left[{\bar{\psi }}_{Z_{+}}^{\left(\xi ,\theta_{+}\right)}\right]}^{I_{+}}$$
(44)$${\bar{P}}_{S}^{\left(\xi \right)}\left(l\right)=\langle p^{\left(\xi \right)}\left(\cdot ,l\right),P_{S}\left(\cdot ,l\right)\rangle$$
(45)$${\bar{\psi }}_{Z_{+}}^{\left(\xi ,\theta_{+}\right)}\left(l_{+}\right)=\langle {\bar{p}}_{+}^{\left(\xi \right)}\left(\cdot ,l_{+}\right),\psi_{Z_{+}}^{\left(\theta_{+}\left(l_{+}\right)\right)}\left(\cdot ,l_{+}\right)\rangle$$
(46)$${\bar{p}}_{+}^{\left(\xi \right)}\left(x_{+},l_{+}\right)=1_{L}\left(l_{+}\right)\frac{\langle P_{S}\left(\cdot ,l_{+}\right)f_{+}\left(x_{+}\left|\cdot \right.,l_{+}\right),p^{\left(\xi \right)}\left(\cdot ,l_{+}\right)\rangle }{{\bar{P}}_{S}^{\left(\xi \right)}\left(l_{+}\right)}\;+\;1_{B_{\;+\;}}\left(l_{+}\right)p_{B,+}\left(x_{+},l_{+}\right)$$
(47)$$p_{Z_{+}}^{\left(\xi ,\theta_{+}\right)}\left(x_{+},l_{+}\right)=\frac{{\bar{p}}_{+}^{\left(\xi \right)}\left(x_{+},l_{+}\right)\psi_{Z_{+}}^{\left(\theta_{+}\left(l_{+}\right)\right)}\left(x_{+},l_{+}\right)}{{\bar{\psi }}_{Z_{+}}^{\left(\xi ,\theta_{+}\right)}\left(l_{+}\right)}.$$

Note that $r_{B,+}\left(l_{+}\right)$ is the birth probability of the target with label $l_{+}$, $p_{B,+}\left(x_{+},l_{+}\right)$ is its kinematic state distribution and $f_{+}\left(x_{+}\left|\cdot \right.,l_{+}\right)$ is the Markov state transition function.

## 5. Experiment

### 5.1. Simulation Environment Settings

We use six sensors consisted of three arrays to observe three acoustic targets as shown in Figure 2. The sensors are at (100 m, 95 m), (95 m, 100 m), (5 m, 100 m), (0 m, 95 m), (0 m, 5 m) and (5 m, 0 m), in [0, 100] × [0, 100] m${}^{2}$.

Three pairs of sensors track the target, each sensor’s observation distance is 150 m, the simulated sound velocity is 344 m/s, the surviving probability is $P_{S}=0.99$ and the clutter intensity of Poisson distribution is $\lambda_{c}=2$. The scenario last 100 s, maximum number of targets is 3. The motion model is a linear state space equation(CV motion model) and the state of target is expressed as:(48)$$X_{k}=AX_{k-1}+B\omega_{k}$$
(49)A=1T000100001T0001B=T2T2220T00T2T2220Tωkt,
where *A* is the target state transition matrix, *B* is the noise matrix and $\omega_{k}$ is process noise and follows a standard Gaussian distribution. The sampling period is T = 1.

In Figure 2, the sensors are displayed by the blue circle, the black circle represents the starting point and the triangle represents the end position. The target location is unknown and two scenarios were compared in this section. The position and velocity vector of target and sensor are represented as $x_{k}={\left[p_{k,x},{\dot{p}}_{k,x},p_{k,y},{\dot{p}}_{k,y}\right]}^{T}$ and $s_{k}^{i}={\left[q_{k,x}^{i},{\dot{q}}_{k,x}^{i},q_{k,y}^{i},{\dot{q}}_{k,y}^{i}\right]}^{T}$, respectively. The range dependent detection probability is defined as:(50)$$P_{D,i}\left(x_{k}\right)\;=\;P_{D,\max}\exp\left(-\frac{{\left[x_{k}-s_{k}^{i}\right]}^{T}C^{T}\Sigma_{D}^{-1}C\left[x_{k}-s_{k}^{i}\right]}{2}\right),$$
where $P_{D,max}=0.98$, $\Sigma_{D}\;=\;\mathrm{diag}{\left(500,500\right)}^{2}$ and $C\;=\;\left[\begin{array}{cccc} 1 & 0 & 0 & 0\\ 0 & 0 & 1 & 0 \end{array}\right]$.

In the scenario 1, We build a parallel model and the survival period of three targets are 1 s–100 s, 10 s–90 s and 20 s–80 s, respectively. The initial states of the three targets are an LMB-RFS with parameters ${\left\{r_{B,k}\left(l_{i}\right),p_{B,k}\left(l_{i}\right)\right\}}_{i=1}^{3}$, where $l_{i}=\left(k,i\right)$, $r_{B,k}\left(l_{i}\right)\;=\;0.02$ and $p_{B}\left(x_{0,i},l_{i}\right)\;=\;N\left(x_{0,i};\mu_{B}^{\left(i\right)};P_{B}\right)$ with:(51)$$\begin{array}{c} \mu_{B}^{\left(1\right)}\;=\;{\left[0\;m,\;1\;m/s,\;90\;m,\;-1\;m/s\right]}^{T}\\ \mu_{B}^{\left(2\right)}\;=\;{\left[0\;m,\;1\;m/s,\;80\;m,\;-1\;m/s\right]}^{T}\\ \mu_{B}^{\left(3\right)}\;=\;{\left[0\;m,\;1\;m/s,\;70\;m,\;-1\;m/s\right]}^{T}\\ P_{B}=diag{\left({\left[0.2,0.08,0.2,0.1\right]}^{T}\right)}^{2}. \end{array}$$

In the scenario 2, the survival period of three targets are 1 s–90 s, 1 s–80 s and 30 s–100 s, respectively. Target 1 and target 2 are born in the same position at the same time. The initial parameters ${\left\{r_{B,k}\left(l_{i}\right),p_{B,k}\left(l_{i}\right)\right\}}_{i=1}^{2}$, $r_{B,k}\left(l_{i}\right)\;=\;0.02$ and $p_{B}\left(x_{0,i},l_{i}\right)\;=\;N\left(x_{0,i};\mu_{B}^{\left(i\right)};P_{B}\right)$ with:(52)$$\begin{array}{c} \mu_{B}^{\left(1\right)}\;=\;{\left[0\;m,\;1\;m/s,\;50\;m,\;0\;m/s\right]}^{T}\\ \mu_{B}^{\left(2\right)}\;=\;{\left[0\;m,\;0.8\;m/s,\;95\;m,\;-0.5\;m/s\right]}^{T}\\ P_{B}=diag{\left({\left[0.2,0.08,0.2,0.1\right]}^{T}\right)}^{2}. \end{array}$$

The experiment uses the three Matlab audio files sample1.wav, sample2.wav and sample3.wav as the acoustic signals of the three targets in the Figure 3.

Taking the acoustic time difference as τ = 0.02 s as an example, the simulation results of the three signals through the cross-correlation algorithm are as shown in the Figure 4.

The time difference of the received signals of sensor arrays are calculated according to the GCC function and the angle difference of each group of sensors is calculated according to the signal receiving direction. The observation equation of the target is defined as:(53)$$z_{k}^{\left[q\right]}=\left[\begin{array}{c} {\hat{\tau }}_{q}\\ \delta_{q} \end{array}\right]+\left[\begin{array}{c} \sigma_{\tau }\\ \sigma_{\delta } \end{array}\right],q=1,\ldots ,Q$$
(54)$${\hat{\tau }}_{q}=\arg\max\int_{-\infty }^{+\infty }\psi_{12}\left(\omega \right)Z_{1}\left(\omega \right)Z_{{}_{2}}^{*}\left(\omega \right)e^{-j\omega \tau_{q}}d\omega$$
(55)$$\delta_{q}\;=\;\left|\arctan\left(\frac{p_{k,y}-q_{k,y}^{1}}{p_{k,x}-q_{k,x}^{1}}\right)-\arctan\left(\frac{p_{k,y}-q_{k,y}^{2}}{p_{k,x}-q_{k,x}^{2}}\right)\right|,$$
where, $z_{k}^{\left[q\right]}$ is nonlinear. At time *k*, ${\hat{\tau }}_{q}$ is the time difference observed by a pair of sensors, $\delta_{q}$ is the angle difference between a pair of sensors receiving signals, $\sigma_{\tau }\;=\;0.001$ s and σδ=ππ720720 rad are the standard deviations of the Gaussian distributed measurement noise. In the scenario 1, three pairs of sensor arrays detected the measurements data of target 1 as show in the Figure 5.

### 5.2. Algorithm Estimation Analysis

#### 5.2.1. Scenario 1

The simulation time is 100 s. The black line in the Figure 6 is the real trajectory of the target and the Red circle and the Color points are the estimated target location. It can be seen from the two pictures that the target tracks obtained from the Gibbs-GLMB filtering and the GLMB filtering are basically consistent with the true trajectory of the targets.

From the simulation results in Figure 6, Figure 7 and Figure 8, it can be seen that the tracking performance of both algorithms is better. In Figure 7 and Figure 8, the cross points are all measurements in the simulation of 1 s–100 s and the points generated outside the target track are false alarms caused by clutter interference. When a target is born, the random clutter may cause false alarms at the position. Nevtheless, in the subsequent tracking and localization, most of these false alarms will be eliminated. Through 100 times Monte Carlo(MC) simulations, the number of targets is estimated as shown in the Figure 9. The red line is algorithm Gibbs-GLMB and the black dotted line is algorithm GLMB. We can see from the comparison of the two algorithms that the number of Gibbs-GLMB estimates is more accurate overall but the estimated number of targets has a large deviation when the actual number of targets changes. The cardinality estimates of targets based on GLMB Filter is always a slightly higher than the true comparison.

We use the Optimal Subpattern Assignment (OSPA) distance [64] (c=100,p=1) to analyze tracking performance. Figure 10 shows the simulation result over 100 MC runs. We can see that OSPA-Loc of two algorithms are very small in the whole process, indicating good estimation performance of the tracker. As shown in the results, the number of targets increases in 0 s, 10 s and 20 s, the OSPA of the GLMB fluctuated, however, the Gibbs-GLMB fluctuated more strongly than GLMB. When the number of targets decreases in 80 s and 90 s, the Gibbs-GLMB results have a large fluctuation, because the three parallel targets are relatively close, the target 3 is false detection; In 87 s–90 s, false cardinality estimates occurs due to the symmetrical geometric relationship between the sensors and the targets. while the GLMB stays a little high but remains relatively stable.

#### 5.2.2. Scenario 2

The target simulation time is 100 s in the Figure 11. The legend in the pictures is the same as that in scenario 1. It also can be seen from the two pictures that the target tracks obtained from the Gibbs-GLMB filtering and the GLMB filtering are basically consistent with true trajectory of the targets.

In the Figure 12, the blue box and the red circle are the estimated position of the particle point and the true position of the target point, respectively. We can see that the targets can be effectively detected through the particles after some steps even the two targets overlap in the same position at beginning.

From the simulation results in Figure 11, Figure 12, Figure 13 and Figure 14, it can be seen that Gibbs-GLMB and GLMB are able to track the target by acoustic and quickly detect the new-born target. But target 3($x_{3}\;=\;{\left[0\;m,\;0.8\;m/s,\;95\;m,\;-0.5\;m/s\right]}^{T}$) with a special starting position which on a sensor position $\left(0\;m,95\;m\right)$, there will be a measurement error and the tracking will be missing detection. Through 100 times Monte Carlo(MC) simulations, the cardinality of targets is estimated as shown in Figure 15. Both algorithms can accurately estimate the cardinality of targets but the cardinality estimates has a large error at the beginning of Gibbs-GLMB. In this scenario, there is no fluctuation when the cardinality of targets changes.

Figure 16 is the OSPA distance over 100 MC runs. The statistical results can further illustrate that the proposed Gibbs-GLMB is more accurate than the GLMB throughout the tracking process, although the error is larger at the beginning of the experiment.

### 5.3. Performance

The filtering algorithms are run in the same PC. The configuration is as follows: CPU: Inter Core MLi5-4200H@2.80 GHz, RAM: 8 GB, using the software matlab2017b, the computation load and accuracy is as described in the following table:

As shown in Table 1, the proposed Gibbs-GLMB filtering is significantly reduced in time-consuming and the cardinality estimates is more accurate compared to the GLMB filtering. The Gibbs-GLMB filtering improves the speed of operation greatly and reduces the complexity of the procedure by integrating updating and prediction of the GLMB filter into one step and by combining the Gibbs sampling algorithm to evaluate the updated target combination, eliminating the target combination with smaller possibility and retaining the target combination with larger weight so the tracking accuracy will be better, and the computational complexity is also significantly reduced due to the reduction of the target combination.

## 6. Conclusions

By introducing the multi-sensor acoustic array and signal detection model, we proposes to use TDOA and AOA measurements, combined with the Gibbs-GLMB filter to track multiple acoustic sources. In this paper, we use RFS theory which can solve the loss of correspondence between set elements with labels and use PHAT combined with the GCC algorithm which improves the result of TDOA calculation through real acoustic signals. The feasibility of the method is verified by tracking multiple nonlinear moving targets. The experimental simulation results show that the Gibbs-GLMB filter can effectively track multi-target but the sensor position will affect the results of the tracking. Compared with the GLMB filter, Gibbs-GLMB filter runs faster and the results are more accurate. The method proposed in this paper is only implemented under ideal simulation conditions. In the future, we will consider applying it to real audio experiments and design an effective sensor array distribution.

## Figures and Tables

**Figure 1 sensors-19-05437-f001:**
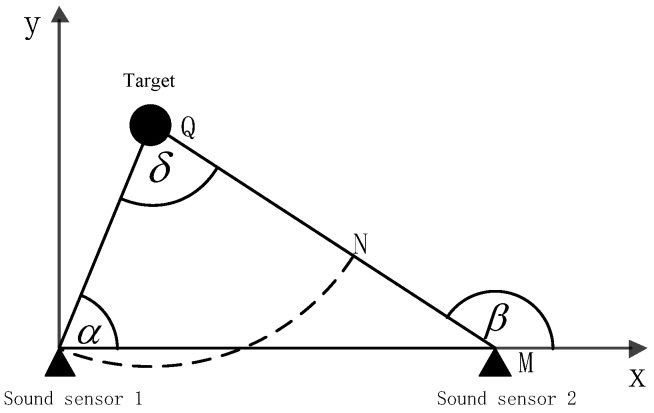
A pair of sensor array model diagrams.

**Figure 2 sensors-19-05437-f002:**
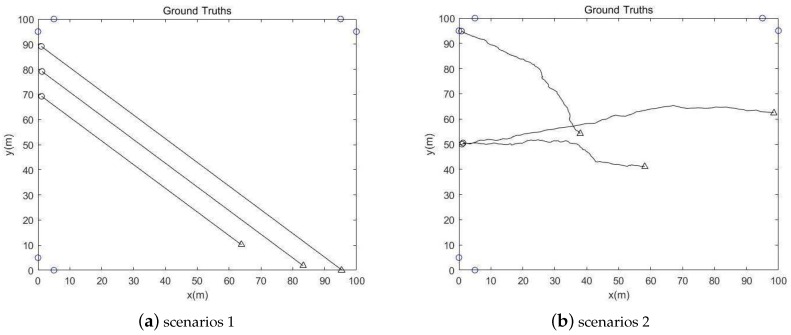
Detection model diagram.

**Figure 3 sensors-19-05437-f003:**
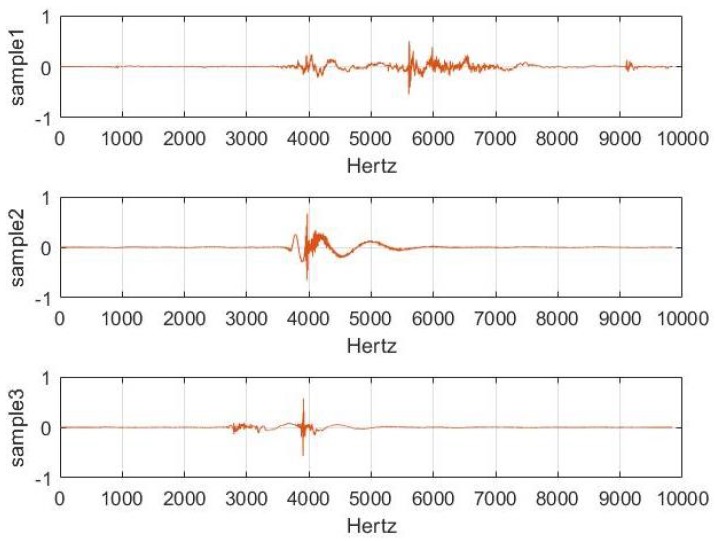
Acoustic signals of the three experimental targets.

**Figure 4 sensors-19-05437-f004:**
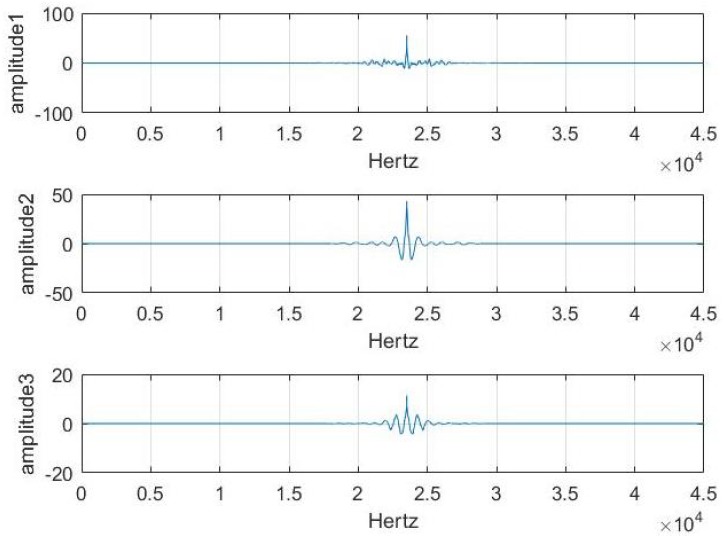
Cross-correlation waveform with a time difference of 0.02 s.

**Figure 5 sensors-19-05437-f005:**
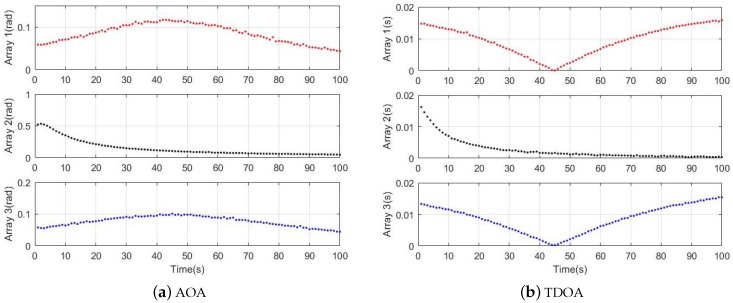
Measurement data.

**Figure 6 sensors-19-05437-f006:**
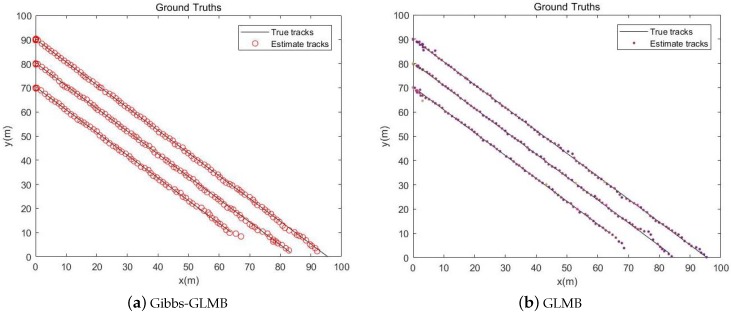
Track estimation.

**Figure 7 sensors-19-05437-f007:**
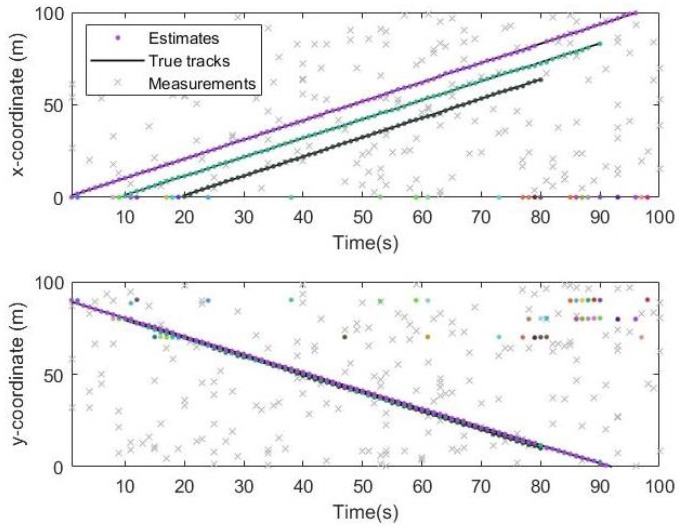
Track result on x and y coordinates by Gibbs-GLMB.

**Figure 8 sensors-19-05437-f008:**
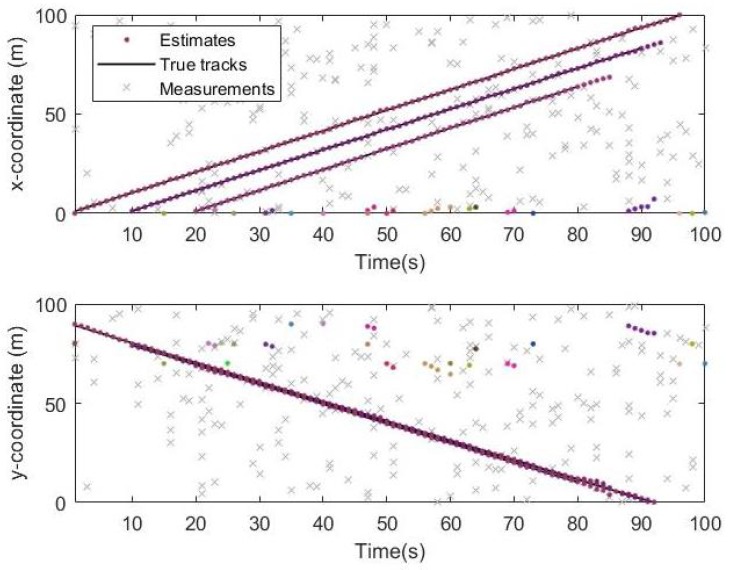
Track result on x and y coordinates by GLMB.

**Figure 9 sensors-19-05437-f009:**
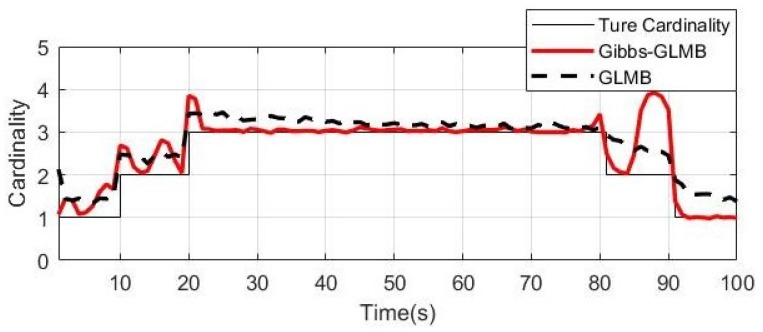
The cardinality estimates (100 times MC).

**Figure 10 sensors-19-05437-f010:**
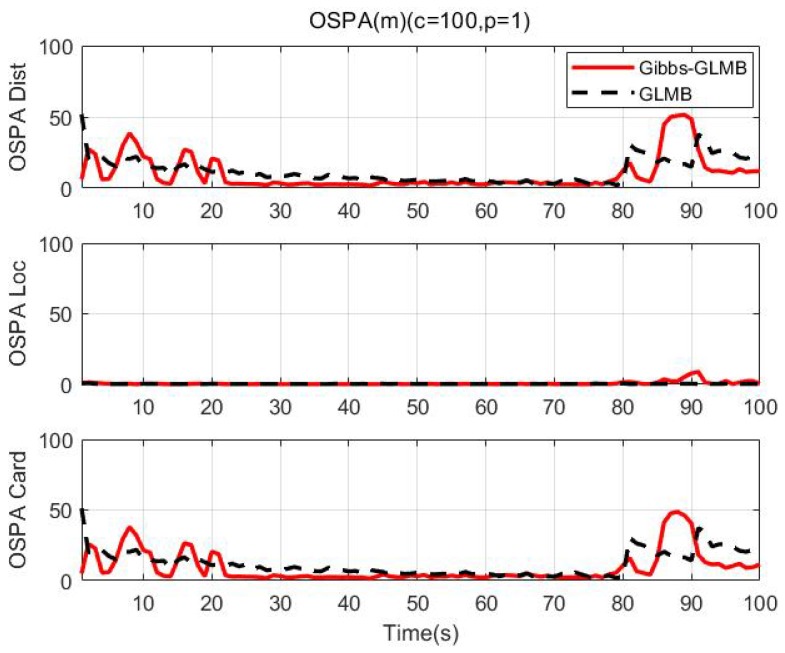
OSPA distance (100 times MC).

**Figure 11 sensors-19-05437-f011:**
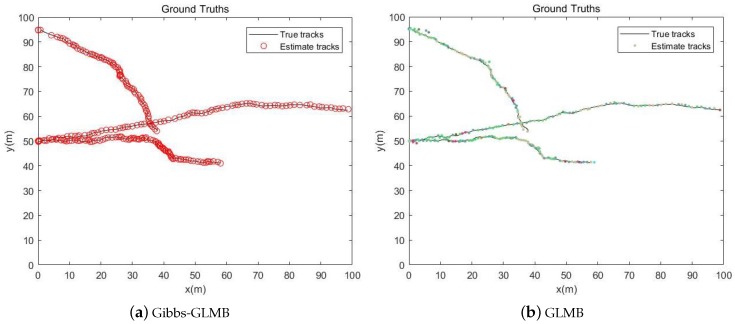
Track estimation.

**Figure 12 sensors-19-05437-f012:**
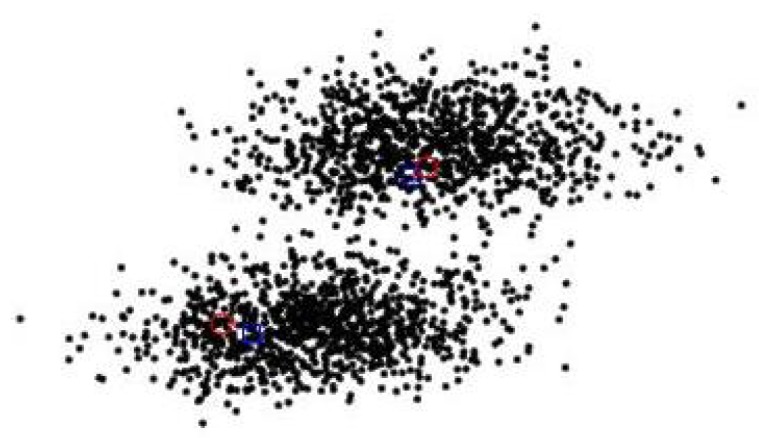
The estimated position of the particle point.

**Figure 13 sensors-19-05437-f013:**
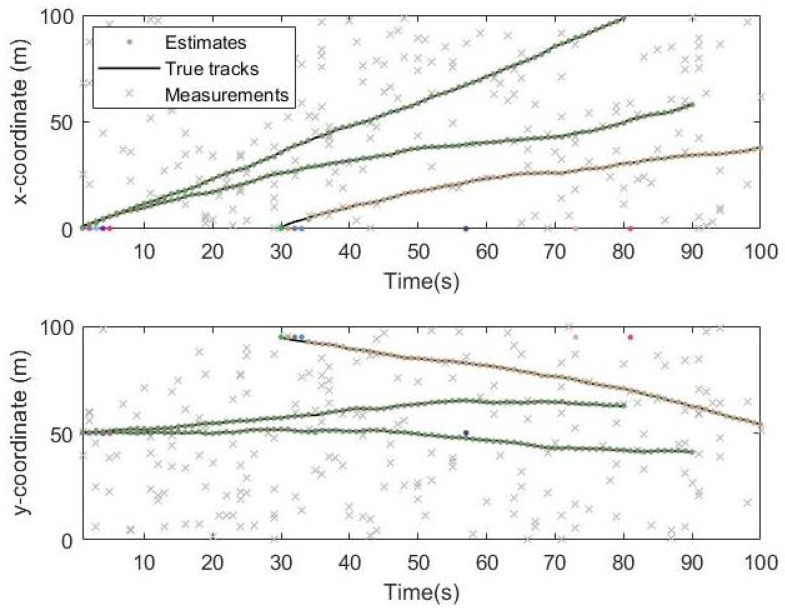
Track result on x and y coordinates by Gibbs-GLMB.

**Figure 14 sensors-19-05437-f014:**
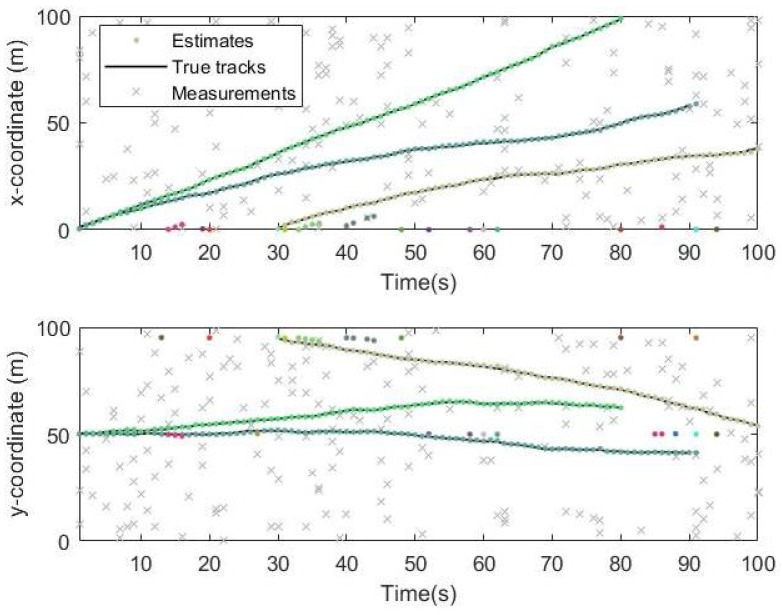
Track result on x and y coordinates by GLMB.

**Figure 15 sensors-19-05437-f015:**
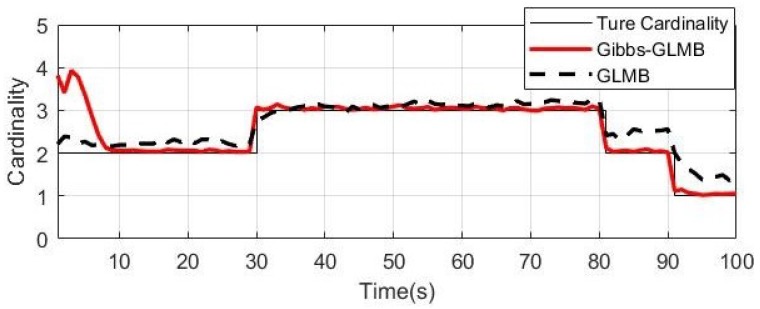
The cardinality estimates (100 times MC).

**Figure 16 sensors-19-05437-f016:**
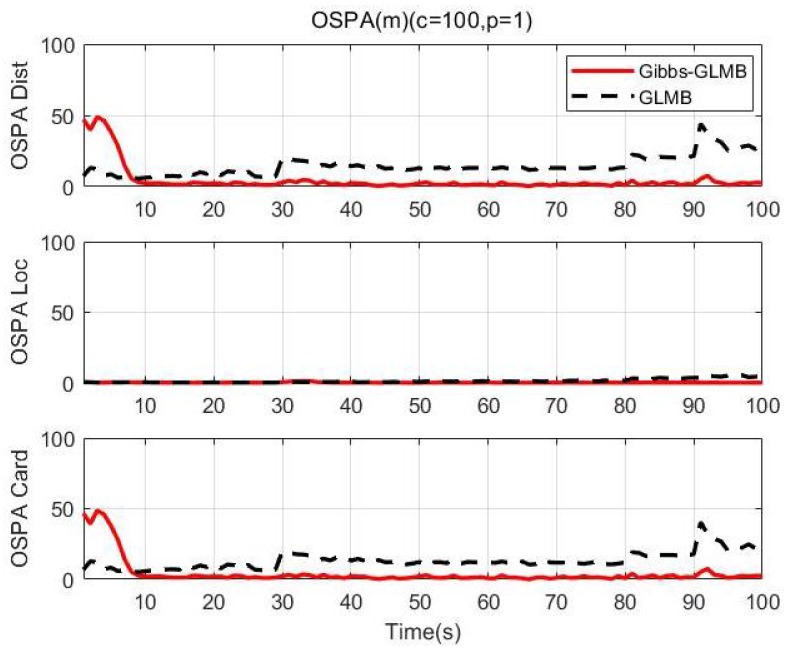
OSPA distance (100 times MC).

**Table 1 sensors-19-05437-t001:** Performance comparison.

Method	Scenario 1		Scenario 2	
	Running Time (s/step)	The Cardinality Accuracy	Running Time (s/step)	The Cardinality Accuracy
Gibbs-GLMB	0.5790	76.86%	0.5950	88.61%
GLMB	1.4188	67.64%	2.1950	61.60%

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
