# Peer review of "Multi-Target Localization and Tracking Using TDOA and AOA Measurements Based on Gibbs-GLMB Filtering†"

_sensors, 2019, doi:10.3390/s19245437_

Round 1

Reviewer 1 Report

In this paper, the authors present a multi-sensor acoustic array model and then track multiple acoustic sources by combining the Gibbs-GLMB filter with the GCC method. The whole paper is well organized but some places need to be improved. My comments are as follows. 1. The motivation for the paper is not very clear. The GLMB filter has been applied to many fields, so the authors should justify why to choose the GLMB filter for passive tracking. Also, compared with traditional methods, what's the challenges in using TDOA and AOA measurements based on the GLMB filter? 2. (1) The transition matrix in Eq.(22) is wrong for the CV model. It should be [1 T;0 1]. (2) There are two 'for' in Lines 1-2 of Algorithm 1. (3) The left-hand side of Eq. (37) should be omega^tilda. 3. The writing should be improved substantially and polished. There are some grammatical errors, for example: (1) the SMC methods is ... in Line 132 should be are, (2) the w_B and w_S are weight of ... in Line 142 should be weights. (3) high-dimensional spaces to low-dimensional one in Line 148 should be space,... (4) the sensors is... in Line 164 should be the sensors are ... 4. Please describe in detail how to implement sampling from the distribution in Line 3 of Algorithm 2? 5. The simulation is insufficient. The authors only show that the proposed Gibbs-GLMB filter is superior to the GLMB filter. But this is because the original Gibbs-GLMB is faster and more accurate than the GLMB filter, just as the authors mentioned in the introduction part. Then, what's the difference between the proposed Gibbs-GLMB filter and the original Gibbs-GLMB filter [34] and their published conference paper [54]? I think it is necessary to supplement the comparison between the proposed method with the traditional passive tracking methods. 6. In the conclusion part, the authors said that they solved the loss of correspondence between set elements with labels. I think this is due to the nature of the labeled RFS.

Author Response

The attachment includes the following two parts:

the first part is the response on pps.1-3 (PDF page number),

the second part is the Revised manuscript on pps.3-25.

Author Response

"The attachment includes the following three parts: the first part is the response on pps.1-4 (PDF page number), the second part is the Revised manuscript on pps.5-26, and the third part is the Ref.[54] on pps. 27-32."

Reviewer 3 Report

Comments on the paper

 ID:sensors-643834

Title: Multi-target localization and tracking using TDOA and AOA measurements based on the Gibbs-GLMB filtering

 This manuscript handles the multi-target tracking problem with TDOA and AOA measurements using fixed acoustic array. A named Gibbs-GLMB filter is proposed for solving the target trajectory and state estimation problem under a cluttered environment. This paper proposes some new insights for multi-target tracking, and this brings information that, for this reviewer, the presented Gibbs-GLMB filter is novel. This reviewer recommends that the paper can be published in the journal after some minor revision.

The comments are below:

 M1: In the modelling of multi-target tracking scenarios, a fixed acoustic array (sensor network) is used. This reviewer suggests the authors to clearly describe how to obtain the observations, TDOA and AOA. Specially, for TDOA measurements in multi-target tracking scenarios, the authors should clearly indicate how to determine this observation is from sensor a and not from sensor b, and then how the concerned TDOA is obtained, since the targets are non-cooperative.

 M2: For the motion model presented in the paper, a CV model is used. However, CV target tracking is a relatively simple issue with respect to other nonlinear motion models. For this reason, this reviewer suggests the authors to conduct nonlinear or hybrid target motion models to test the proposed Gibbs-GLMB filter.

 M3: In the scenarios, a fixed acoustic array is used. It is well known that the localization and tracking performance is heavily depends on the sensor-target geometry. The authors are encouraged to explain this concern with the proposed Gibbs-GLMB filter for the considered multi-target localization and tracking.

 M4: This reviewer observes that there are some grammar mistakes or typos at the last two paragraphs in the introduction section. The reviewer suggests the authors to do a double pass to the whole text.

 M5: In Eq.(15), the function “arctan()” is used as arctan(y/x), not as arctan((x,y)). Some explanations or rewritten works are needed.

 M6: Some reference formats are needed to be fixed in the revised version.

Author Response

Please see the attachment.
The attachment includes the following two parts:
the first part is the response on pps.1-4 (PDF page number),
the second part is the Revised manuscript on pps.5-26.

Reviewer 4 Report

This paper proposes a multi-target localization and tracking method using Gibbs generalized label multi-Bernoulli (Gibbs-GLMB) filter. Comparing with the GLMB method, the new method adopts Gibbs sampling algorithm such that it is more accurate and significantly saves computing time (2 ~ 3 times faster). 

This paper presents a useful method in target tracking. It is recommended for publication.

Author Response

Please see the attachment.
The attachment includes the following two parts:
the first part is the response on pps.1 (PDF page number),
the second part is the Revised manuscript on pps.2-23.

Round 2

Reviewer 1 Report

The authors have supplied motivation and provided more sufficient simulations. All the issues I cared about are revised and most grammatical errors have been corrected. Therefore, it is suggested to accept the paper.     
    However, some references should be cited in the paper,for instance, the labeled multi-Bernoulli distribution on page 2, the OSPA metric in the simulation.  

Author Response

Thanks to the reviewer for reminding me, I have added relevant references, as well as references that supplemented a few formulas, which are marked in the attachment, See pps.1,2,4,16.

Reviewer 2 Report

I feel the authors have addressed most of the comments, which results in the manuscript being much improved compared to the previous version. Thus, I believe this manuscript is ready for possible publications.   

Author Response

Thank you for your support and positive!